# Chronic Endometritis and Uterine Endometrium Microbiota in Recurrent Implantation Failure and Recurrent Pregnancy Loss

**DOI:** 10.3390/biomedicines11092391

**Published:** 2023-08-27

**Authors:** Kanako Takimoto, Hideto Yamada, Shigeki Shimada, Yoshiyuki Fukushi, Shinichiro Wada

**Affiliations:** 1Department of Obstetrics and Gynecology, Teine Keijinkai Hospital, Sapporo 006-8555, Japan; takimoto-ka@keijinkai.or.jp (K.T.); yfukushi294ks@keijinkai.or.jp (Y.F.); 6200@keijinkai.or.jp (S.W.); 2Center for Recurrent Pregnancy Loss, Teine Keijinkai Hospital, Sapporo 006-8555, Japan; 3Department of Obstetrics and Gynecology, Mommy’s Clinic Chitose, Chitose 066-0038, Japan; shimashige@hotmail.co.jp

**Keywords:** chronic endometritis, microbiota, recurrent pregnancy loss, repeated implantation failure, uterine endometrium

## Abstract

The aim of this study was to evaluate whether chronic endometritis (CE) and uterine endometrium microbiota were associated with repeated implantation failures (RIFs) and recurrent pregnancy losses (RPLs). In this prospective study, uterine endometrial specimens were obtained from 24 women with RIF, 27 with RPL, and 29 fertile control women. Immunohistochemical staining of CD138 for CE and 16S ribosomal RNA (rRNA) sequencing analysis for uterine endometrium microbiota were performed simultaneously. To assess CE, Liu’s method, McQueen scores and plasma cell count/10 mm^2^ were used. The frequency of CE (plasma cells > 5.15/10 mm^2^) was higher in women with RPL (29.6%) than in fertile controls (6.8%, *p* < 0.05). The plasma cell count/10 mm^2^ in women with RPL (median 1.53, range 0–252.6, *p* < 0.01) and women with RIF (median 0.6, range 0–6.98, *p* < 0.05) was higher than in fertile controls (median 0, range 0–29). The uterine endometrium microbiota in women with RPL or RIF was not significantly different from that in fertile controls. However, the relative dominance rate of *Lactobacillus iners* (median 4.7%, range 0–99.9 vs. median 0%, range 0–100, *p* < 0.001) and the positive rate of *Ureaplasma* species (36.3% vs. 8.6%, *p* < 0.05) were higher in 11 women with CE than in 69 women without CE. The results suggest that CE may be involved in the pathophysiology of RPL and RIF. *Lactobacillus iners* and *Ureaplasma* species may be associated with the etiology of CE.

## 1. Introduction

Chronic endometritis (CE) is normally histologically diagnosed as plasma cells’ infiltration into the uterine endometrial stroma, although universal criteria for the CE diagnosis have not been determined [1,2]. Some investigators have shown possible adverse effects of CE on human reproduction [3,4,5]. The frequency of CE is 2.8–56.8% in infertility, 14–67.5% in recurrent implantation failure (RIF), and 9.3–67.6% in recurrent pregnancy loss (RPL) [6]. McQeen et al. [7] reported that 56% (60/107) of women with RPL had CE, and the live birth rate was higher in RPL women without CE (87.1%, 27/31) than in those with untreated CE (67.6%, 23/34), but without a statistical significance (*p* = 0.08). Immunohistochemistry of the plasma cell marker CD138 (syndecan-1) is a more reliable method than Hematoxylin-Eosin staining with respect to plasma cell detection, and it can be used clinically to diagnose CE [6,7].

Recently, microbiome analyses with 16S ribosomal RNA (rRNA) analysis using a next-generation sequencer have become popular in reproductive techniques [8,9]. Moreno et al. [10] first evaluated the uterine endometrium microbiota in infertility using 16S rRNA sequence analysis, and found that rates of implantation, pregnancy, and live birth in in-vitro fertilization and embryo transfer (IVF-ET) were higher in women with *Lactobacillus*-dominant microbiota (>90%) than in women with non-*Lactobacillus*-dominant microbiota. Shi et al. [11] also evaluated the vaginal microbiota in women with threatened premature labor using 16S rRNA sequence analysis, and found that increases in *Ureaplasma* species and decreases in *Lactobacillus* species were associated with subsequent preterm delivery in a cohort study.

To understand whether CE or an abnormality of uterine endometrium microbiota is involved in the pathophysiology of RIF and RPL, this prospective study assessed uterine endometrium in women with RIF, women with RPL, and fertile control women by histopathological analyses for CE together with 16S rRNA sequence analyses for uterine endometrium microbiota. Bacterial species of *Lactobacillus*, *Ureaplasma*, *Mycoplasma*, *Gardnerella*, *Prevotella*, *Streptococcus*, *Atopobium*, *Dialister*, *Bifidobacterium*, *Anaerococcus*, *Escherichia*, and *Enterococcus*, which are possibly associated with CE, were examined. A relationship between CE and uterine endometrium microbiota was also evaluated.

## 2. Materials and Methods

### 2.1. Study Participants

This prospective cohort study was approved by Teine Keijinkai Hospital Ethics Committee. Informed consent was obtained from all participants. Between March 2021 and January 2023, 24 women with two or more repeated implantation failures (RIF), 27 women with two or more recurrent pregnancy losses (RPL), and 29 fertile control women were enrolled. The fertile control women were under 44 years old and had regular menstrual cycles for more than one year and a history of at least one normal delivery without a history of infertility, RPL, uterine myoma, adenomyosis, endometriosis, malignancy, or surgery requiring intrauterine manipulation after the last delivery. In women with RIF, one had endometriosis, and two had polycystic ovary syndrome (PCOS). In women with RIF or RPL, none had uterine myoma. Only four women with RIF had received antibiotics treatments two or more menstrual cycles before an endometrial biopsy. In women with RIF or RPL, none had received probiotics/prebiotics.

### 2.2. Procedures

An endometrial biopsy was performed during the mid-luteal phase, confirmed by transvaginal ultrasound and the last menstrual cycle. The vaginal wall and perineum were washed with 0.025 *w*/*v*% benzalkonium chloride solution and wiped with a clean cotton swab, and a sampling pipette (Pipet CuretTM, CooperSurgical, Inc., Trumbull, CT, USA) was then inserted into the uterus. Endometrial specimens were collected by aspiration. One-third of the specimen before and after each aspiration tube was immersed in 8 mL of 10% neutral buffered formalin solution and examined for histopathology, including immunohistochemical staining of CD138. Specimens in the middle portion of the aspiration tube were immersed in a container kit OMNIgene^®^-VAGINAL for microbiome (DNA Genotek Inc., Ottawa, ON, Canada) containing DNA/RNA stabilizers. The samples were immediately transferred to Varinos Inc., Tokyo, Japan, where uterine endometrial microbiota were analyzed using the 16S rRNA sequence method. The variable region 4 (V4), the hypervariable region of the 16S rRNA gene, was amplified by polymerase chain reaction (PCR) using DNA extracted from tissue specimens [10]. An amplified PCR sample was identified according to the Illumina 16S Metagenomic Sequencing Library Preparation protocol [12].

Histopathological analyses for CE were performed at SAPPORO CLINICAL LABORATORY INC., Sapporo, Japan, using immunohistochemical staining for CD138. CD138-positive cells in uterine endometrium were defined as plasma cells. CE was diagnosed when the plasma cell count was >5.15/10 mm^2^, according to Liu’s method [13]. CE was also assessed by DB McQueen scores: 0 = none, <1 plasma cell/HPF (×40); 1 = 1–5/HPF or clusters of <20 cells; 2 = 6–20/HPF or clusters of >20 cells; and 3 ≧ 20/HPF or sheets of cells [7]. McQeen et al. [7] first reported an association between CE and the live birth rate in women with RPL. Liu et al. [13] first demonstrated an association between CE and uterine endometrial microbiota in infertile women. Therefore, these methods, as well as plasma cell count/10 mm^2^, were used for the CE assessment in this study.

### 2.3. Statistical Analysis

Clinical characteristics and backgrounds, plasma cell count/10 mm^2^ with CD138 staining, the frequency of CE (Liu’s method), the McQueen DB score, and uterine endometrium microbiota were compared between fertile control women and women with RPL or with RIF. Uterine endometrium microbiota were compared between women with and women without CE.

The EZR (Saitama Medical Center, Jichi Medical University, Saitama, Japan), a graphical user interface of R (The R Foundation for Statistical Computing, Vienna, Austria), was used for statistical analyses in this study [14]. Categorical variables were compared by Fisher’s exact test and Mann–Whitney *U* test. All *p* values were two-sided, with *p* values less than 0.05 considered statistically significant.

## 3. Results

Table 1 shows a comparison of clinical characteristics and backgrounds between fertile control women and women with RPL or with RIF. There were no differences in the body mass index, whereas the age, gravidity, parity, number of previous miscarriages, and number of implantation failures significantly differed between the two groups.

Table 2 shows a comparison of CE and uterine endometrium microbiota between fertile control women and women with RPL or with RIF. In a comparison of CE, there were no differences in the McQueen DB score between fertile controls and women with RPL or with RIF. The CD138-positive plasma cell count/10 mm^2^ was significantly higher in women with RPL (median 1.53, range 0–252.6, *p* < 0.01) and women with RIF (median 0.6, range 0–6.98, *p* < 0.05) than in fertile controls (median 0, range 0–29). The frequency of CE as diagnosed by Liu’s method (plasma cell count > 5.15/10 mm^2^) was significantly higher in women with RPL (29.6%) than in fertile controls (6.8%, *p* < 0.05). However, there were no differences in the number of bacterial species, the relative dominance rate of *Lactobacillus* species, or the frequency of *Lactobacillus*-dominant microbiota, which was defined as >90% of the relative dominance rate of *Lactobacillus* species. The positive rate of *Lactobacillus*, *Ureaplasma*, *Mycoplasma*, *Gardnerella*, *Prevotella*, *Streptococcus*, *Atopobium*, *Dialister*, *Bifidobacterium*, *Anaerococcus*, *Escherichia*, or *Enterococcus* species was not different between fertile control women and women with RPL or with RIF (Table 2).

Table 3 shows a comparison of uterine endometrium microbiota between 11 women with and 69 women without CE, as diagnosed by Liu’s method. The 11 women with CE (plasma cell count > 5.15/10 mm^2^) included 1 woman with RIF, 8 women with RPL, and 2 fertile control women, while the 69 women without CE included 23 women with RIF, 19 women with RPL, and 27 fertile control women. The relative dominance rate of *Lactobacillus iners* in women with CE (median 4.7%, range 0–99.9%) was significantly higher (*p* < 0.001) than in women without CE (median 0%, range 0–100%). The presence of *Ureaplasma* species in women with CE (4/11, 36.3%) was significantly higher (*p* < 0.05) than in women without CE (6/69, 8.6%). There were no differences in the number of bacterial species, the relative dominance rates of other *Lactobacillus* species, or the frequency of *Lactobacillus*-dominant microbiota (>90%). The positive rate of *Lactobacillus*, *Mycoplasma*, *Gardnerella*, *Prevotella*, *Streptococcus*, *Atopobium*, *Dialister*, *Bifidobacterium*, *Anaerococcus*, *Escherichia*, or *Enterococcus* species was not different between women with CE and without CE (Table 3).

## 4. Discussion

In this prospective study, histopathological analyses for CE and 16S rRNA sequence analyses for uterine endometrium microbiota were performed simultaneously on the uterine endometrium obtained at the mid-luteal phase in women with RPL, women with RIF, and fertile control women. This study demonstrated for the first time that frequencies of CE in women with RPL, as well as the plasma cell counts in women with RPL and women with RIF, were higher than those in fertile control women. These results suggest that CE may be involved in the pathophysiology of RPL and RIF. Women with CE had a higher relative dominance rate of *Lactobacillus iners* and a higher positive rate of *Ureaplasma* species compared with women without CE. Therefore, these microorganisms may be associated with the etiology of CE.

McQeen et al. [7] first reported that 56% of women with RPL had CE, and the live birth rate was higher in RPL women without CE than in those with CE, but without a statistical significance. A prospective cohort study of women with RPL or RIF demonstrated that increased plasma cell counts in the uterine endometrium were associated with miscarriage in subsequent pregnancies [15]. The clinical pregnancy rate of IVF-ET in infertile women with CE was lower than in women without CE [16]. Concerning associations between treatments for CE and pregnancy outcomes, antibiotics treatment for CE before IVF-ET in women with RIF improved the live birth rate [17] and the pregnancy rate [18]. However, a randomized controlled trial found that antibiotics treatment for women with reproductive failure had no efficacy on the pregnancy rate or the miscarriage rate [19]. In the present study, CE was associated with RPL, and increased plasma cell counts in the uterine endometrium were associated with RPL and RIF. Local chronic inflammation at the uterine endometrium may be causally associated with RPL and RIF. A cohort study to assess the efficacy of treatment with antibiotics and probiotics/prebiotics on pregnancy outcomes in women with RPL and women with RIF is ongoing.

Concerning the association between CE and uterine endometrium microbiota, Liu et al. [13] reported that 9% of infertile women had CE (plasma cell count > 5.15/10 mm^2^) and that the relative dominance rates of *Lactobacillus* species determined by 16S rRNA sequence analyses were only 1.9% in infertile women with CE and 81% in infertile women without CE. The present study examined uterine endometrial specimens, but not intrauterine fluids, and found that the rates of *Lactobacillus* species were not different between women with CE (98.4%) and women without CE (96.5%), whereas positive rates of *Lactobacillus iners* and *Ureaplasma* species were higher in women with CE than in women without CE. *Mycoplasma*, *Gardnerella*, *Prevotella*, *Streptococcus*, *Atopobium*, *Dialister*, *Bifidobacterium*, *Anaerococcus*, *Escherichia*, and *Enterococcus* species in the uterine endometrium are causally associated with CE [6,13,19]. In the present study, however, positive rates of these bacterial species were not different between women with RPL or RIF and fertile control women, or between women with CE and women without CE.

Moreno et al. [10] first evaluated the uterine endometrium microbiota in infertility using 16S rRNA sequence analysis, and found that rates of implantation, pregnancy, and live birth in IVF-ET were higher in women with *Lactobacillus*-dominant microbiota (>90%) than in women with non-*Lactobacillus*-dominant microbiota. However, studies using 16S rRNA sequence analyses demonstrated that pregnancy rates of IVF-ET in infertile women were not associated with uterine endometrium microbiota [20,21]. There was no difference in pregnancy or miscarriage rates in IVF-ET between women with *Lactobacillus*-dominant microbiota (>80%) and women with non-*Lactobacillus*-dominant microbiota [22]. A study found that the relative dominance rates of *Lactobacillus* species in uterine endometrium microbiota were not different between women with RIF and control women [23], and another study also showed that the rates of *Lactobacillus* species were not different between infertile women with RIF and those with non-RIF [24]. The results of these studies suggest that the relative dominance rates of *Lactobacillus* species are not associated with pregnancy outcomes in IVF-ET or RIF. Similarly, in the present study, uterine endometrium microbiota in women with RPL or RIF were not different from those in fertile controls. However, a recent prospective study using 16S rRNA sequence analyses demonstrated that an increase in *Ureaplasma* species in the uterine endometrium microbiota led to a risk of miscarriage of a fetus with a normal chromosome karyotype and preterm delivery in subsequent pregnancies in women with RPL [25]. *Ureaplasma* species cause uterine infection, chorioamnionitis, as well as adverse pregnancy outcomes [26]. In a cohort study, *Mycoplasma genitalium*, *Mycoplasma hominis*, *Ureplasma parvum*, and *Ureplasma urealyticum* were detected by PCR-based methods at frequencies of 0.8%, 11.2%, 52.0%, and 8.7%, respectively, in the vagina of 877 women during early pregnancy [27]. They found that the presence of *Ureplasma urealyticum* was determined as a risk factor for late miscarriage and preterm delivery. 16S rRNA sequence analyses for uterine endometrium microbiota in women with RPL can identify microbiota associated with adverse pregnancy outcomes, and treatment with antibiotics and probiotics/prebiotics before conception may improve subsequent pregnancy outcomes.

The question of whether *Lactobacillus iners* are beneficial or pathogenic to the host microbiome is controversial [28]. Several investigators suggest that vaginal *Lactobacillus crispatus* is beneficial to pregnancy, while *Lactobacillus iners* may be adverse [28,29,30]. The present study found for the first time that *Lactobacillus iners* was more frequently detected in women with CE than in women without CE, although the median relative dominance rate of *Lactobacillus iners* in women with CE was only 4.7% (range 0–99.9%). Therefore, not only *Lactobacillus iners,* but other microorganisms associated with the presence of *Lactobacillus iners,* may cause pathological CE.

The results of the present study suggest that CE may be involved in the pathophysiology of RPL and RIF and that *Lactobacillus iners* and *Ureaplasma* species may be associated with the etiology of CE. Treatments with antibiotics and probiotics/prebiotics before conception may improve subsequent pregnancy outcomes in these women with CE. The results of this study provide useful information for clinical practitioners who investigate the etiologies and risk factors for RPL and RIF. However, the number of study participants is not enough. Further studies are needed to confirm our results. It is also necessary to compare uterine endometrium microbiota between healthy women and women with gynecological disorders, such as PCOS, endometriosis, and uterine leiomyoma.

## 5. Conclusions

This study simultaneously performed histopathological analyses for CE and 16S rRNA sequence analyses for uterine endometrium microbiota on uterine endometrium specimens, but not on intrauterine fluids, and demonstrated for the first time that the frequencies of CE in women with RPL (29.6% vs. 6.8%) and the plasma cell count/10 mm^2^ in women with RPL (median 1.53 vs. median 0) and women with RIF (median 0.6 vs. median 0) were higher than in fertile control women. The relative dominance rate of *Lactobacillus iners* (median 4.7% vs. median 0%) and the positive rates of *Ureaplasma* species (36.3% vs. 8.6%) were higher in women with CE than in women without CE. The CE may be involved in the pathophysiology of RPL and RIF. *Lactobacillus iners* and *Ureaplasma* species may be associated with the etiology of CE.

## Figures and Tables

**Table 1 biomedicines-11-02391-t001:** Clinical characteristics and backgrounds.

Factors	Repeated Implantation Failure *n* = 24	*p* Value	Recurrent Pregnancy Loss *n* = 27	*p* Value	Control *n* = 29
Age, years	40 (32–45)	<0.001	37 (23–45)	0.053	36 (27–43)
Body mass index, kg/m^2^	21.3 (16.4–34.7)	0.41	22.6 (18.0-28.8)	0.67	21.2 (16.9–37.0)
Gravidity	2 (0–10)	0.218	3 (2–10)	<0.001	2 (1–3)
Parity	0.5 (0–2)	<0.001	0 (0–2)	<0.001	2 (1–3)
Number of previous miscarriages	1 (0–8)	<0.01	3 (2–8)	<0.001	0 (0–1)
Number of implantation failures	3.5 (2–7)	<0.001	0 (0–7)	<0.01	0 (0–0)

Median (range); Mann–Whitney *U* test.

**Table 2 biomedicines-11-02391-t002:** Chronic endometritis and uterine endometrium microbiota.

Chronic Endometritis	Repeated Implantation Failure *n* = 24	*p* Value	Recurrent Pregnancy Loss *n* = 27	*p* Value	Control *n* = 29
Plasma cell count/10 mm^2^ with CD138 staining	0.60 (0–6.98)	<0.05	1.53 (0–252.6)	<0.01	0 (0–29)
Plasma cell count > 5.15/10 mm^2^ (Liu’s method)	1 (4.1%)	1	8 (29.6%)	<0.05	2 (6.8%)
McQueen DB score					
Score 0	7		6		21
Score 1	17		18		8
Score 2	0		2		0
Score 3	0		1		0
McQueen DB score ≧ 2	0 (0%)	NA	3 (11.1%)	0.106	0 (0%)
**Uterine endometrium microbiota**					
Number of bacterial species	3 (2–16)	0.949	4 (1–17)	0.88	3 (1–17)
Relative dominance rate of *Lactobacillus* species, %	87.9 (0–99.9)	0.157	97.0 (0–100)	0.774	97.2 (0–100)
*Lactobacillus*-dominant microbiota (>90%)	11 (45.8%)	0.277	15 (55.5%)	0.786	18 (62.0%)
Presence of *Lactobacillus* species	19 (79.1%)	0.715	26 (96.2%)	0.353	25 (86.2%)
Relative dominance rate of *Lactobacillus crispatus*, %	0 (0–99.7%)	0.576	0 (0–99.9%)	0.63	0 (0–99.8%)
Relative dominance rate of *Lactobacillus gasseri*, %	0 (0–99.7%)	0.706	0 (0–86.7%)	0.723	0 (0–97.1%)
Relative dominance rate of *Lactobacillus iners*, %	0 (0–98.6%)	0.085	0 (0–99.9%)	0.349	0.1 (0–100%)
Relative dominance rate of *Lactobacillus jensenii*, %	0 (0–98.3%)	0.636	0 (0–89.3%)	0.882	0 (0–15.5%)
Presence of *Ureaplasma* species	0 (0%)	0.117	6 (22.2%)	0.497	4 (13.7%)
Presence of *Mycoplasma* species	0 (0%)	NA	1 (3.7%)	0.482	0 (0%)
Presence of *Gardnerella* species	7 (29.1%)	1	11 (40.7%)	0.399	8 (27.5%)
Presence of *Prevotella* species	10 (41.6%)	0.384	9 (33.3%)	0.773	8 (27.5%)
Presence of *Streptococcus* species	9 (37.5%)	0.372	3 (11.1%)	0.299	7 (24.1%)
Presence of *Atopobium* species	3 (12.5%)	0.318	8 (29.6%)	0.765	7 (24.1%)
Presence of *Dialister* species	4 (16.6%)	1	7 (25.9%)	0.523	5 (17.2%)
Presence of *Bifidobacterium* species	8 (33.3%)	0.212	4 (14.8%)	1	5 (17.2%)
Presence of *Anaerococcus* species	2 (8.3%)	0.584	2 (7.4%)	0.605	1 (3.4%)
Presence of *Escherichia* species	1 (4.1%)	0.453	0 (0%)	NA	0 (0%)
Presence of *Enterococcus* species	0 (0%)	NA	0 (0%)	NA	0 (0%)

Median (range); Mann–Whitney *U* test, Fisher’s exact test. McQueen DB score: 0, none <1 plasma cell/HPF (×40); 1, 1–5/HPF or clusters of <20 cells. 2, 6–20/HPF or clusters of >20 cells; 3, >20/HPF or sheets of cells.

**Table 3 biomedicines-11-02391-t003:** Comparison of uterine endometrium microbiota between women with and without chronic endometritis.

Uterine Endometrium Microbiota	Women with Chronic Endometritis (Liu’s Method) *n* = 11	Women without Chronic Endometritis (Liu’s Method) *n* = 69	*p* Value
Number of bacterial species	6 (2–17)	3 (1–17)	0.543
Relative dominance rate of *Lactobacillus* species, %	98.4 (0.1–99.9)	96.5 (0–100)	0.839
*Lactobacillus*-dominant microbiota (>90%)	7 (63.6%)	37 (53.6%)	0.746
Presence of *Lactobacillus* species	11 (100%)	59 (85.5%)	0.342
Relative dominance rate of *Lactobacillus crispatus, %*	0 (0–94.5)	0 (0–99.9)	0.423
Relative dominance rate of *Lactobacillus gasseri*, %	0 (0–50.5)	0 (0–99.7)	0.231
Relative dominance rate of *Lactobacillus iners*, %	4.7 (0–99.9)	0 (0–100)	<0.001
Relative dominance rate of *Lactobacillus jensenii*, %	0 (0–1.4)	0 (0–98.3)	0.154
Presence of *Ureaplasma* species	4 (36.3%)	6 (8.6%)	<0.05
Presence of *Mycoplasma* species	1 (9.0%)	0 (0%)	0.137
Presence of *Gardnerella* species	5 (45.4%)	21 (30.4%)	0.324
Presence of *Prevotella* species	4 (36.3%)	23 (33.3%)	1
Presence of *Streptococcus* species	1 (9.0%)	18 (26.0%)	0.445
Presence of *Atopobium* species	4 (36.3%)	14 (20.2%)	0.256
Presence of *Dialister* species	3 (27.2%)	13 (18.8%)	0.685
Presence of *Bifidobacterium* species	1 (9.0%)	16 (23.1%)	0.441
Presence of *Anaerococcus* species	1 (9.0%)	4 (5.7%)	0.533
Presence of *Escherichia* species	0 (0%)	1 (1.4%)	1
Presence of *Enterococcus* species	0 (0%)	0 (0%)	NA

Median (range); Mann–Whitney *U* test, Fisher’s exact test.

## Data Availability

Not applicable.

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
