# Peer review of "Chronic Endometritis and Uterine Endometrium Microbiota in Recurrent Implantation Failure and Recurrent Pregnancy Loss"

_biomedicines, 2023, doi:10.3390/biomedicines11092391_

Round 1
Reviewer 1 Report
I appreciate the opportunity to review the manuscript entitled “Chronic endometritis and uterine endometrium microbiota in recurrent implantation failure and recurrent pregnancy loss” submitted in journal Biomedicines.
The aim of the presented work was to evaluate whether chronic endometritis and uterine endometrium microbiota are associated with repeated implantation failures and recurrent pregnancy losses.
Although the overall impression about the manuscript is good, I would suggest some changes outlined below:
1. Please provide a list of abbreviations used in manuscript.
2. Please add data on McQueen DB score and Liu’s method in M&M section with the corresponding references and explain why the authors decided to use these two particular methods to evaluate their patients.
3. Please extend the discussion. Authors should provide explanations related to the possible mechanisms of how endometrial microbiome influences the implantation failure and recurrent pregnancy loss.
4. Is there any difference / role of the endometrial microbiome between healthy population and women diagnosed with the different gynecological disorders such as PCOS, endometriosis, uterine leiomyoma etc.
5. Please compare the results observed in your work with the results observed by other authors about the influence of the endometrial microbiome on fertility as well as the miscarriage.
6. Are there the probiotics and/or other treatment options which could be used in the future for the treatment of the infertile women affected with recurrent implantation failure and those with recurrent pregnancy loss (other than suggested antibiotics)?
7. Shortcomings and strong points of the study should be explained in more details.
Considering the abovementioned comments this submission meets the criteria to be published in journal Biomedicine after major revisions.
Author Response
Answers to comments
Reviewer 1
- Please provide a list of abbreviations used in manuscript.
According to reviewers’ suggestion, in Page 8, Line 271
“A list of abbreviations: CE, chronic endometritis; IVF-ET, in-vitro fertilization and embryo transfer; RPL, recurrent pregnancy losses; RIF, repeated implantation failures; rRNA, ribosomal RNA”
was added.
- Please add data on McQueen DB score and Liu’s method in M&M section with the corresponding references and explain why the authors decided to use these two particular methods to evaluate their patients.
According to reviewers’ suggestion, in Page 2, Line 95 in M&M
“CE was diagnosed when plasma cell count was >5.15/10 mm2 [9]. CE was also assessed by DB McQueen scores: 0=none, <1 plasma cell/HPF (×40); 1=1-5/HPF or clusters of <20 cells; 2=6-20/HPF or clusters of >20 cells: and 3≧20/HPF or sheets of cells [4].“
was changed to
“CE was diagnosed when plasma cell count was >5.15/10 mm2 according to Liu’s method [9]. CE was also assessed by DB McQueen scores: 0=none, <1 plasma cell/HPF (×40); 1=1-5/HPF or clusters of <20 cells; 2=6-20/HPF or clusters of >20 cells: and 3≧20/HPF or sheets of cells [4]. In this study, to assess CE, Liu’s method, DB McQueen scores and plasma cell count/10 mm2 were used.”
And in Page 1, Line 17
“and plasma cell count/10 mm2”
was inserted.
- Please extend the discussion. Authors should provide explanations related to the possible mechanisms of how endometrial microbiome influences the implantation failure and recurrent pregnancy loss.
The present study found that the uterine endometrium microbiota was not associated with RPL or RIF as compared with fertile controls, but CE was associated with RPL and RIF.
According to reviewers’ suggestion, in Page 6, Line 184
“Local chronic inflammation at the uterine endometrium may be causally associated with RPL and RIF.”
was added.
- Is there any difference / role of the endometrial microbiome between healthy population and women diagnosed with the different gynecological disorders such as PCOS, endometriosis, uterine leiomyoma etc.
Thank you for important questions.
According to reviewers’ suggestion, in Page 2, Line 71
“One had endometriosis, and two had polycystic ovary syndrome (PCOS) in women with RIF. None had uterine myoma in women with RIF or RPL.”
was added.
And in Page 7, Line 243
“It is also necessary to compare uterine endometrium microbiota between healthy women and women with gynecological disorders such as PCOS, endometriosis, and uterine leiomyoma.”
was added.
- Please compare the results observed in your work with the results observed by other authors about the influence of the endometrial microbiome on fertility as well as the miscarriage.
According to reviewers’ suggestion, in Page 6, Line 195
“Mycoplasma, Gardnerella, Prevotella, Streptococcus, Atopobium, Dialister, Bifidobacterium, Anaerococcus, Escherichia, and Enterococcus species in the uterine endometrium are causally associated with CE [3, 9, 15]. In the present study, however, positive rates of these bacterial species were not different between women with RPL or RIF and fertile control women, or between women with CE and women without CE.”
was added.
- Are there the probiotics and/or other treatment options which could be used in the future for the treatment of the infertile women affected with recurrent implantation failure and those with recurrent pregnancy loss (other than suggested antibiotics)?
According to reviewers’ suggestion, in Page 6, Line 186
“The cohort study to assess efficacy of antibiotics treatment on pregnancy outcome in women with RPL and women with RIF is ongoing.”
was changed to
“The cohort study to assess efficacy of treatment with antibiotics and probiotics/prebiotics on pregnancy outcome in women with RPL and women with RIF is ongoing.”
According to reviewers’ suggestion, in Page 7, Line 224
“The 16S rRNA sequence analyses for uterine endometrium microbiota in women with RPL can identify microbiota associated with adverse pregnancy outcomes, and antibiotic treatments before conception may improve subsequent pregnancy outcomes.”
was changed to
“The 16S rRNA sequence analyses for uterine endometrium microbiota in women with RPL can identify microbiota associated with adverse pregnancy outcomes, and treatment with antibiotics and probiotics/prebiotics before conception may improve subsequent pregnancy outcomes.”
- Shortcomings and strong points of the study should be explained in more details.
According to reviewers’ suggestion, in Page 7, Line 248
“This study performed histopathological analyses for CE and 16S rRNA sequence analyses for uterine endometrium microbiota simultaneously on the uterine endometrium specimens but not on intrauterine fluids, and for the first time demonstrated that frequencies of CE in women with RPL, and the plasma cell counts in women with RPL and women with RIF were higher than those in fertile control women. The positive rates of Lactobacillus iners and Ureaplasmaspecies were higher in women with CE than in women without CE.”
was added.

Reviewer 2 Report
In the manuscript „Chronic endometritis and uterine endometrium microbiotain recurrent implantation failure and recurrent pregnancy loss” the authors try to evaluate whether chronic endometriosis and uterine endometrium microbiota were associated with recurrent implantation failure and recurrent pregnancy loss. The study has a relevant theme, however, the paper needs major revision before being accepted for publication.
The article does not explain how the assessed bacteria influence the pathogenesis of CE.
No specific arguments verify the hypothesis. Conclusions do not result from the conducted research and require redrafting.
My questions are:
The composition of the microbiome in the female reproductive system depends on many factors, including medications taken, previous antibiotic therapy, etc.
Has this information been included?
Is bacterial supplementation included? Many patients use it alone.
Abstract
Line 12-23: Please standardize the way of presenting the results. Sometimes the authors give the median and range, other times only the range
Introduction
Line 32-33: What do the authors mean by plasma cells? Please explain.
There is no information about the role of the microbiome, the importance of individual bacteria and why they were selected. Please complete the introduction.
Why did the authors choose CD138 for evaluation? What is the role of the CD138 molecule? Please complete the introduction.
Results
Median 0 at range (0 - 99.7%) raises doubts (Table 2 and 3)
Please recalculate the results.
The P-Value should be written in italics and should be present as:
P<0.05, P<0.01, P<0.001, P<0.0001 or P>0.05
Please, correct this in all manuscript.
Discussion
Line 158-159: What could be the relationship?
Line 178-179: How can it be related?
Line 208-219: What do the authors think is the impact?
What specific information is provided by the results of the authors' research?
Conclusion
Please redraft the conclusions so that they result from the conducted research
Author Response
Reviewer 2
My questions are:
- The composition of the microbiome in the female reproductive system depends on many factors, including medications taken, previous antibiotic therapy, etc. Has this information been included? Is bacterial supplementation included? Many patients use it alone.
Thank you for important questions.
According to reviewers’ suggestion, in Page 2, Line 73
“Only four women with RIF had received antibiotics treatments two or more menstrual cycles before an endometrial biopsy. None had received probiotics/prebiotics in women with RIF or RPL.”
- Abstract
Line 12-23: Please standardize the way of presenting the results. Sometimes the authors give the median and range, other times only the range.
According to reviewers’ suggestion, in Page 1, Line 19
“The plasma cell count/10 mm2 in women with RPL (median 1.53, range 0-252.6, p<0.01) and women with RIF (0.6, 0-6.98, p<0.05) was higher than that in fertile controls (0, 0-29). The uterine endometrium microbiota in women with RPL or RIF was not significantly different from that in fertile controls. However, the relative dominance rate of Lactobacillus iners (median 4.7%, range 0-99.9 vs. 0%, 0-100, p<0.001) and the positive rate of Ureaplasma species (36.3% vs. 8.6%, p<0.05) were higher in 11 women with CE than those in 69 women without CE.”
was corrected to
“The plasma cell count/10 mm2 in women with RPL (median 1.53, range 0-252.6, P <0.01) and women with RIF (median0.6, range 0-6.98, P <0.05) was higher than that in fertile controls (median 0, range 0-29). The uterine endometrium microbiota in women with RPL or RIF was not significantly different from that in fertile controls. However, the relative dominance rate of Lactobacillus iners (median 4.7%, range 0-99.9 vs. median 0%, range 0-100, P <0.001) and the positive rate of Ureaplasma species (36.3% vs. 8.6%, P <0.05) were higher in 11 women with CE than those in 69 women without CE.”
- Introduction
Line 32-33: What do the authors mean by plasma cells? Please explain.
According to reviewers’ suggestion, in Page 1, Line 41
“Immunohistochemistry of the plasma cell marker CD138 (syndecan-1) is a more reliable method than Hematoxylin-Eosin staining with respect to plasma cell detection, and can be used clinically to diagnose CE. [3, 4]”
was added.
- There is no information about the role of the microbiome, the importance of individual bacteria and why they were selected. Please complete the introduction.
Thank you for proper suggestions.
In addition to Gardnerella species, species of Prevotella, Streptococcus, Atopobium, Dialister, Bifidobacterium, Anaerococcus, Escherichia, and Enterococcus, which are associated with CE, were examined. These results are shown in Table 2 and Table 3.
Introduction in Page 2, Line 57
“Bacterial species of Lactobacillus, Ureaplasma, Mycoplasma, Gardnerella, Prevotella, Streptococcus, Atopobium, Dialister, Bifidobacterium, Anaerococcus, Escherichia, and Enterococcus, which are possibly associated with CE, were examined.”
was added.
And in Page 3, Line 130
“However, there were no differences in the number of bacterial species, relative dominance rate of Lactobacillus species, Lactobacillus dominant microbiota which was defined as >90% of the relative dominance rate of Lactobacillus species, presence of Lactobacillus species, Gardnerella species, Ureaplasma species, or Mycoplasma species
between fertile control women and women with RPL or with RIF (Table 2)”.
was changed to
“However, there were no differences in the number of bacterial species, relative dominance rate of Lactobacillus species, or Lactobacillus dominant microbiota which was defined as >90% of the relative dominance rate of Lactobacillus species. The positive rate of Lactobacillus, Ureaplasma, Mycoplasma, Gardnerella, Prevotella, Streptococcus, Atopobium, Dialister, Bifidobacterium, Anaerococcus, Escherichia, or Enterococcus species was not different between fertile control women and women with RPL or with RIF (Table 2).”
And in Page 5, Line 151
“There were no differences in the number of bacterial species, relative dominance rates of other Lactobacillus species, a frequency of Lactobacillus dominant microbiota (>90%), presence of Lactobacillus species, Gardnerella species, or Mycoplasma species, between women with CE and without CE.”
was changed to
“There were no differences in the number of bacterial species, relative dominance rates of other Lactobacillus species, or a frequency of Lactobacillus dominant microbiota (>90%). The positive rate of Lactobacillus, Mycoplasma, Gardnerella, Prevotella, Streptococcus, Atopobium, Dialister, Bifidobacterium, Anaerococcus, Escherichia, or Enterococcus species was not different between women with CE and without CE. (Table 3)”
- Why did the authors choose CD138 for evaluation? What is the role of the CD138 molecule? Please complete the introduction.
According to reviewers’ suggestion, in Page 1, Line 41
“Immunohistochemistry of the plasma cell marker CD138 (syndecan-1) is a more reliable method than Hematoxylin-Eosin staining with respect to plasma cell detection, and can be used clinically to diagnose CE. [3,4]”
was added.
- Results
Median 0 at range (0 - 99.7%) raises doubts (Table 2 and 3). Please recalculate the results.
The relative dominance rates of Lactobacillus crispatus in 24 women with RIF in Table 2 are as follows; 0, 0, 0, 0, 0, 0, 0, 0, 0, 0, 0, 0, 0, 0, 0, 0.1, 0.1, 0.2, 50.8, 84.7, 86.9, 99.5, 99.6, 99.7%, resulting in median 0, range 0-99.7%, for example. These percentage numbers are not normal probability distribution, so that mean±SD could not be used in statistical analyses.
- The P-Value should be written in italics and should be present as:
P<0.05, P<0.01, P<0.001, P<0.0001 or P>0.05
Please, correct this in all manuscript.
Thank you for proper suggestions. P-Value was corrected in tables and the manuscript.
- Discussion
Line 158-159: What could be the relationship?
According to reviewers’ suggestion, in Page 7, Line 169
“Although uterine endometrium microbiota in women with RPL or RIF was not significantly different from that in fertile controls, women with CE had a higher relative dominance rate of Lactobacillus iners and a higher positive rate of Ureaplasma species compared with women without CE, suggesting that these microorganisms may be associated with the etiology of CE.”
was changed to
“Women with CE had a higher relative dominance rate of Lactobacillus iners and a higher positive rate of Ureaplasmaspecies compared with women without CE. Therefore, these microorganisms may be associated with the etiology of CE.”
- Line 178-179: How can it be related?
According to reviewers’ suggestion, in Page 6, Line 192
“In the present study, rates of Lactobacillus species were not different between women with CE (98.4%) and women without CE (96.5%), although Lactobacillus iners and Ureaplasma species were associated with the presence of CE.”
was changed to
“The present study examined the uterine endometrial specimens but not intrauterine fluids, and found that rates of Lactobacillus species were not different between women with CE (98.4%) and women without CE (96.5%). However, positive rates of Lactobacillus iners and Ureaplasma species were higher in women with CE than in women without CE.”
- Line 208-219: What do the authors think is the impact?
What specific information is provided by the results of the authors' research?
According to reviewers’ suggestion, in Page 7, Line 237
“Results of the present study suggest that CE may be involved in the pathophysiology RPL and RIF, and thatLactobacillus iners and Ureaplasma species may be associated with the etiology of CE. The treatments with antibiotics and probiotics/prebiotics before conception may improve subsequent pregnancy outcomes in these women with CE.”
was added.
- Conclusion
Please redraft the conclusions so that they result from the conducted research
According to reviewers’ suggestion, in Page 8, Line 247
“This study performed histopathological analyses for CE and 16S rRNA sequence analyses for uterine endometrium microbiota simultaneously on the uterine endometrium specimens but not on intrauterine fluids, and for the first time demonstrated that frequencies of CE in women with RPL, and the plasma cell counts in women with RPL and women with RIF were higher than those in fertile control women. The positive rates of Lactobacillus iners and Ureaplasmaspecies were higher in women with CE than in women without CE.”
was added.

Round 2
Reviewer 1 Report
The manuscript was significantly approved by the authors.
Still, it requires minor revision .
It would be nice if the authors explained why they chose McQueen DB score and Liu’s method, and not some other (i.e., there are no other scores/methods, or these were considered the best for reasons which should be explained).
Secondly, it would be an asset if shortcomings and strong points of the study paragraph contains more data.
Minor editing of English language required.
Author Response
Answers to comments
Reviewer 1
The manuscript was significantly approved by the authors.
Still, it requires minor revision.
- It would be nice if the authors explained why they chose McQueen DB score and Liu’s method, and not some other (i.e., there are no other scores/methods, or these were considered the best for reasons which should be explained).
According to reviewers’ suggestion, in Page 3, Line 99
“In this study, to assess CE, Liu’s method, DB McQueen scores and plasma cell count/10 mm2 were used.”
was changed to
“McQeen et al. [7] first reported an association between CE and live birth rate in women with RPL. Liu et al. [13] first demonstrated an association between CE and uterine endometrial microbiota in infertile women. Therefore, these methods as well as plasma cell count/10 mm2 were used for CE assessment in this study.”
- Secondly, it would be an asset if shortcomings and strong points of the study paragraph contains more data.
According to reviewers’ suggestion, in Page 7, Line 250
“This study performed histopathological analyses for CE and 16S rRNA sequence analyses for uterine endometrium microbiota simultaneously on the uterine endometrium specimens but not on intrauterine fluids, and for the first time demonstrated that frequencies of CE in women with RPL, and the plasma cell counts in women with RPL and women with RIF were higher than those in fertile control women. The positive rates of Lactobacillus iners and Ureaplasmaspecies were higher in women with CE than in women without CE.”
was changed to
“This study performed histopathological analyses for CE and 16S rRNA sequence analyses for uterine endometrium microbiota simultaneously on the uterine endometrium specimens but not on intrauterine fluids, and for the first time demonstrated that frequencies of CE in women with RPL (29.6% vs. 6.8%), and the plasma cell count/10 mm2 in women with RPL (median 1.53 vs. median 0) and women with RIF (median 0.6 vs. median 0) were higher than those in fertile control women. The relative dominance rate of Lactobacillus iners (median 4.7% vs. median 0%) and the positive rates ofUreaplasma species (36.3% vs. 8.6%) were higher in women with CE than in women without CE.”
